# Health-related quality of life variations by sociodemographic factors and chronic conditions in three metropolitan cities of South Asia: the CARRS study

Kavita Singh,[1,2,3,4] Dimple Kondal,[1,2,3] Roopa Shivashankar,[1,2,3]
Mohammed K Ali,[1,5,6] Rajendra Pradeepa,[7] Vamadevan S Ajay,[1,2,3]
Viswanathan Mohan,[7] Muhammad M Kadir,[8] Mark Daniel Sullivan,[9] Nikhil Tandon,[1,4]
K M Venkat Narayan,[1,5,6] Dorairaj Prabhakaran[1,2,3,10]

## ABSTRACT

**Objectives** Health-related quality of life (HRQOL) is a key indicator of health. However, HRQOL data from representative populations in South Asia are lacking. This study aims to describe HRQOL overall, by age, gender and socioeconomic status, and examine the associations between selected chronic conditions and HRQOL in adults from three urban cities in South Asia.

**Methods** We used data from 16 287 adults aged ≥20 years from the baseline survey of the Centre for Cardiometabolic Risk Reduction in South Asia cohort (2010–2011). HRQOL was measured using the European Quality of Life Five Dimension—Visual Analogue Scale (EQ5D-VAS), which measures health status on a scale of 0 (worst health status) to 100 (best possible health status).

**Results** 16 284 participants completed the EQ5D-VAS. Mean age was 42.4 (±13.3) years and 52.4% were women. 14% of the respondents reported problems in mobility and pain/discomfort domains. Mean VAS score was 74 (95% CI 73.7 to 74.2). Significantly lower health status was found in elderly (64.1), women (71.6), unemployed (68.4), less educated (71.2) and low-income group (73.4). Individuals with chronic conditions reported worse health status than those without (67.4 vs 76.2): prevalence ratio, 1.8 (95% CI 1.61 to 2.04).

**Conclusions** Our data demonstrate significantly lower HRQOL in key demographic groups and those with chronic conditions, which is consistent with previous studies. These data provide insights on inequalities in population health status, and potentially reveal unmet needs in the community to guide health policies.

## Strengths and limitations of this study

► This is the first population-level health-related quality of life (HRQOL) data from South Asia using the European Quality of Life Five Dimension—Visual Analogue Scale including three large metropolitan cities in India and Pakistan with a large sample size (16 284 adults aged ≥20 years).

► Our data provide the first baseline values to be used for monitoring population health status and analysed the relationships between selected chronic conditions and HRQOL.

► HRQOL data presented in this article could be used to complement national health targets by providing a measure of chronic disease burden based on perceived health status rather than solely on mortality and disease prevalence.

► Due to the cross-sectional nature of the data, the causal relationship between socioeconomic parameters or chronic conditions and HRQOL cannot be determined.

► Many chronic conditions (respiratory, locomotor, cancer and others) were not included in the survey. Therefore, the ranking of most severe health conditions and associated HRQOL is not complete.

For numbered affiliations see end of article.

**Correspondence to**
Dr Dorairaj Prabhakaran;
dprabhakaran@ccdcindia.org

## INTRODUCTION

Health-related quality of life (HRQOL) is a multidimensional concept that provides a broader perspective of health through conveying an individual's ability to function in physical, mental and social domains of life.[1] HRQOL is thus an essential patient-centred outcome measure that is useful to guide health policies.[2,3] HRQOL is preferred over other health indicators (life expectancy, mortality, morbidity) for measuring chronic disease burden as it incorporates both length and quality of life.[4] In South Asia, chronic conditions (diabetes, hypertension and heart diseases) occur at an early age,[5] with detrimental effects on length and quality of life.[6,7] In addition, episodes and fear of hypoglycaemia, recurrent heart attacks, stroke and other long-term complications (kidney diseases, diabetic retinopathy) are not always measured as such, although they have a substantial adverse impact on an individual's overall health status.[8] Therefore, it is important to quantify the effect of chronic conditions on individuals' HRQOL.

There are several disease-specific (Chronic Respiratory Distress Questionnaire, Arthritis Impact Measurement Scale) and generic instruments (Short Form 36, WHO—Quality of Life Questionnaire and the European Quality of Life Five Dimension—Visual Analogue Scale (EQ5D-VAS)) available to measure population HRQOL.[4 9–18] However, the EQ5D-VAS is favoured because it is generic, not specific to a particular disease, and it includes multidimensional measures of health profile in five dimensions (mobility, self-care, usual activities, pain/discomfort and anxiety/depression) and the single-dimensional measure VAS to summarise overall health status.[1] Also, EQ5D-VAS has been applied and validated for its use in many population surveys across the world; therefore, it makes the comparison of health status across populations easier.

Data on population HRQOL across socioeconomic status (SES) from South Asia are scarce, and little is known about the relative associations between different chronic conditions and individual HRQOL. The Centre for Cardiometabolic Risk Reduction in South Asia (CARRS) study[19] had collected data on both EQ5D-VAS and selected chronic conditions from a large representative population of adults in urban South Asia. We used this opportunity to examine population HRQOL in this region. In this paper, we describe the variations in HRQOL by age, gender and SES, and explore the relationships between selected chronic conditions and HRQOL in a representative sample of adults aged ≥20 years from three metropolitan cities in India and Pakistan. We also analysed the relationship between multidimensional EQ5D measures and single-dimensional VAS across major subgroups.

## METHODS
### Study design and setting
We obtained data from the baseline cross-sectional survey of the CARRS cohort (2010–2011), which recruited a representative sample of non-pregnant adults aged ≥20 years from three urban cities: Chennai, Delhi and Karachi. These metropolitan cities with large and heterogeneous populations in terms of demographic profile and economic transitions offer unique opportunities to assess variations in health status across different socioeconomic groups. The detailed CARRS study design has been published elsewhere.[19] Briefly, a multistage cluster random sampling strategy was used with wards (in Delhi and Chennai) or clusters (in Karachi) as the primary sampling units. Using the WHO STEPS (STEPwise approach to surveillance) survey 'Kish method', two participants, one male and one female, aged ≥20 years (non-pregnant) and meeting the study eligibility criteria, were selected from each randomly selected household.[19]

### Study measures
Comprehensive and uniform data collection instruments were used to capture measurements in all three sites. A summary of all surveillance measures, methods and instruments used in the study has been published in detail.[19] Briefly, a questionnaire was administered to collect information regarding demographic, socioeconomic, behavioural, and past and present health status of the participant.

Trained study staff measured anthropometric parameters (height, weight) using standardised techniques and blood pressure (BP) twice at each participant's home or at a medical camp organised in the community, after 5 min in a seated position using an electronic BP measuring device (Omron Dalian, Liaoning Sheng, China). If the difference between the first two systolic or diastolic BP readings was more than 10 mm Hg or 5 mm Hg, respectively, a third reading was taken. Average BP readings of the two/three readings were recorded in the study database. Additionally, fasting blood glucose (FBG) and glycated haemoglobin (HbA1c) were measured. The overall response rates were 94.7% for questionnaire completion and 84.3% for blood tests.

Population health status was measured using the EQ5D-VAS questionnaire, which consisted of two components: health state description and self-rated health status on VAS. Health state description (profile) includes five dimensions (5D): mobility (walking ability), self-care (ability to wash or dress by oneself), usual activities (ability to work, study, housework), pain/discomfort and anxiety/depression. The respondents self-rate their level of severity for each dimension using three levels (EQ5D-3L): having no problems, having some or moderate problems, or being unable to do/having extreme problems. The respondents were asked to choose one of the statements that best described their health status on the surveyed day. For example, three levels of 'mobility' dimension were phrased as 'I have no problems in walking', 'I have some problems in walking' and 'I am confined to bed'. Given the possible permutations of different domains and response types, there are potentially 243 (=$3^5$) different health profiles.

For overall health status, the respondents evaluated their health status using the VAS. The VAS asks respondents to mark health status on the day of the interview on a scale of 0 (worst health status) to 100 (best imaginable health status).

### Covariates
Self-reported age at baseline in completed years was used and categorised into 20–24, 25–34, 35–44, 45–54, 55–64, 65–74 and ≥75. Based on participant responses, we categorised employment status into employed, student, housewife, retired and unemployed. Income class was grouped into three categories based on household monthly income: low-income, less than 10 000 Indian rupees (equivalent to US$200); middle-income, 10 000–20 000 Indian rupees (US$200–400); and high-income strata, greater than 20 000 Indian rupees (US$400). We categorised highest education level attained into three categories: up to primary, secondary schooling and graduates. The marital status was classified as single, married, widowed and divorced. Body

mass index (kg/m$^2$) international classification of ≤17.9 was used to define underweight, 18.0–24.9=normal weight, 25.0–29.9=overweight and ≥30.0=obese. Lifestyle habits like tobacco use were classified based on self-reports as never, former and current user. Data on chronic conditions consisted of self-reported hypertension, diabetes, heart disease, stroke and kidney disease. In addition, diabetes was categorised into self-reported, newly diagnosed (defined by no self-reported diabetes and FBG of ≥126 mg/dL or HbA1c ≥6.5%), pre-diabetes (no self-reported diabetes and FBG ≥100–125 mg/dL or HbA1c ≥5.7%–6.4%) and normo-glycaemia (no self-reported diabetes and FBG <100 mg/dL and HbA1c <5.7%). Similarly, we classified hypertension as self-reported, newly diagnosed (no self-reported hypertension and BP ≥140/90 mm Hg), prehypertension (no self-reported hypertension and BP 120–139/80–89 mm Hg) and normotensive (no history of hypertension and BP <120/80 mm Hg).

### Ethical considerations
Study participants provided written informed consent before participation in the study.

### Analysis
We used Stata V.14.0 for data analysis. We used the 'svy' command for all analysis to account for the complex survey design.[20] Before any of the survey estimation commands were used, the svyset command was used to specify the variables that described the stratification, sampling weight and primary sampling unit variables. This analysis included data obtained from 16 284 study participants. All the responses coded as refused, unknown or missing were treated as missing data. The frequency (percentages) and mean were reported to display the level of population health status and the sample characteristics. Percentages of those reporting any problems in EQ5D domains and mean VAS were stratified by respondents' demographic characteristics—age, gender, marital status and SES—education, income and employment status; and health-related indicators—presence of chronic conditions—were reported. Additionally, prevalence ratios of moderate or severe health problems in people with and without chronic conditions were estimated using log binomial regression. Where the model did not reach convergence, Poisson regression model was used. The model was adjusted for sociodemographic covariates (age, gender, marital status, education level and household income) and city. Linear regression analysis was performed to explore the relationship between the VAS and the EQ5D measures across major subgroups. In the regression model, VAS was used as a dependent variable, and EQ5D measures were treated as independent variables.

### STUDY RESULTS
### Characteristics of the study population
A total of 17 274 individuals in 10 002 households were approached in the three study sites (7596 participants in Chennai, 5420 in Delhi, 4258 in Karachi). From

these, a total of 16 287 participants were recruited (the overall response rate was 94.3% at the participant level: 6906 Chennai (90.9%), 5364 Delhi (98.9%) and 4017 Karachi (94.3%)). Detailed baseline characteristics of the CARRS cohort are published elsewhere.[21–24] Briefly, the mean age was 42.4 (±13.3), 52.4% were female, 61% completed secondary schooling and the majority of respondents (72.5%) reported household income level <10 000 Indian rupees (US$200). Two-thirds (66%) of the study population had BMI ≥25, one-fifth (20%) of the respondents reported current tobacco use, and 37.5% had self-reported chronic conditions (hypertension, diabetes, heart disease, stroke or chronic kidney disease).

### Overall HRQOL by age and gender
A total of 16 284 study participants completed the EQ5D-VAS (99.9%). Overall, the percentage of respondents reporting any problems in mobility and pain/discomfort (14% each) was higher than for other domains. Greater health problems were observed with higher age for both men and women (p<0.001) (table 1). Problems with mobility were higher with advancing age. However, problems with anxiety/depression did not show such trend. Average health status (VAS) reported by the CARRS cohort was 74.5 (95% CI 73.7 to 74.2) (figure 1). Women reported lower health status than men (71.6 vs 79.0; p<0.001).

Of the respondents 74% rated a perfect health profile with no difficulties in any EQ-5D domain, and 0.06% rated the worst health profile whereby they had difficulties with every EQ-5D domain. The distribution of the VAS scores was skewed in the direction of best-imagined health state. Only 0.5% respondents rated their health status on VAS under 10, and 10% rated it under 50 (online supplementary appendix 1).

### HRQOL and SES
Table 2 and Figure 2 depict the mean VAS, percentage and prevalence ratios of respondents reporting moderate or severe problems in the five dimensions, across various subgroups, respectively. Employed adults and students reported better health status than home makers, retired or unemployed participants. We observed almost equal health status in home makers and retired people. Health status was also similar in the middle-income and high-income groups, while it was significantly lower in the low-income group. Individuals with higher education (graduate and above) and high income had higher HRQOL than those with secondary or primary schooling and low-income class. Also, individuals with BMI ≥18–24 kg/m$^2$ reported better health status than those with BMI ≥25 kg/m$^2$. Current tobacco users reported better health status than former tobacco users or non-users. However, in a stratified analysis of HRQOL in tobacco users by presence or absence of chronic conditions, tobacco users with chronic conditions reported worse health status than non-users.

**Table 1** Percentage of respondents reporting moderate or severe problems in EQ5D domains, stratified by age and gender

| EQ5D dimensions | 20–24 years | 25–34 years | 35–44 years | 45–54 years | 55–64 years | 65–74 years | ≥75 years | Overall |
|---|---|---|---|---|---|---|---|---|
| Overall (N) | 1179 | 3752 | 4672 | 3539 | 2005 | 878 | 262 | 16287 |
| Male (N) | 591 | 1614 | 2128 | 1723 | 1026 | 500 | 178 | 7760 |
| Female (N) | 588 | 2138 | 2544 | 1816 | 979 | 378 | 84 | 8527 |
| **Mobility** | | | | | | | | |
| All respondents (%) | 5.3 | 8.3 | 13.6 | 18.1 | 23.5 | 31.2 | 39 | 14.6 |
| 95% CI | [4.0 to 7.0] | [6.8 to 10.0] | [11.7 to 15.9] | [16.1 to 20.3] | [20.8 to 26.4] | [27.5 to 35.2] | [32.7 to 45.6] | [13.3 to 15.9] |
| Male (%) | 2.9 | 3.6 | 6 | 8.7 | 17 | 20.5 | 34.7 | 8.2 |
| 95% CI | [1.5 to 5.6] | [2.6 to 5.0] | [4.8 to 7.3] | [7.2 to 10.4] | [14.1 to 20.4] | [16.7 to 24.9] | [27.5 to 42.7] | [7.3 to 9.2] |
| Female (%) | 7.8 | 11.8 | 20.1 | 26.9 | 30.6 | 45.6 | 48 | 20.3 |
| 95% CI | [5.9 to 10.2] | [9.7 to 14.4] | [17.1 to 23.4] | [24.0 to 29.9] | [26.5 to 35.1] | [40.8 to 50.6] | [36.2 to 60.1] | [18.5 to 22.3] |
| **Self-care** | | | | | | | | |
| All respondents (%) | 1.6 | 2.6 | 3.8 | 4.7 | 6.9 | 9 | 14.6 | 4.2 |
| 95% CI | [1.0 to 2.8] | [2.0 to 3.5] | [2.9 to 5.0] | [3.8 to 5.7] | [5.4 to 8.8] | [6.9 to 11.8] | [10.3 to 20.2] | [3.6 to 4.9] |
| Male (%) | 1.2 | 1.5 | 1.7 | 2.7 | 5.1 | 5.2 | 14 | 2.6 |
| 95% CI | [0.4 to 3.6] | [0.8 to 2.5] | [1.0 to 2.9] | [1.8 to 3.9] | [3.1 to 8.2] | [3.3 to 8.0] | [8.9 to 21.5] | [2.0 to 3.3] |
| Female (%) | 2.1 | 3.5 | 5.5 | 6.5 | 8.9 | 14.2 | 15.6 | 5.6 |
| 95% CI | [1.2 to 3.5] | [2.5 to 5.0] | [4.1 to 7.5] | [5.2 to 8.2] | [6.7 to 11.6] | [10.6 to 18.7] | [9.4 to 24.9] | [4.6 6.8] |
| **Usual activities** | | | | | | | | |
| All respondents (%) | 2 | 3.5 | 4.8 | 7.1 | 10.7 | 16.6 | 23.1 | 6.0 |
| 95% CI | [1.2 to 3.2] | [2.8 to 4.4] | [3.8 to 5.9] | [6.0 to 8.4] | [8.9 to 12.8] | [13.9 to 19.7] | [17.8 to 29.4] | [5.4 to 6.8] |
| Male (%) | 1.3 | 1.4 | 1.8 | 3.2 | 6.4 | 11 | 19.7 | 3.2 |
| 95% CI | [0.5 to 3.7] | [0.9 to 2.1] | [1.2 to 2.7] | [2.3 to 4.3] | [4.8 to 8.6] | [7.9 to 15.2] | [13.8 to 27.2] | [2.7 to 3.9] |
| Female (%) | 2.6 | 5.1 | 7.2 | 10.7 | 15.4 | 24.1 | 30.3 | 8.5 |
| 95% CI | [1.6 to 4.3] | [4.0 to 6.6] | [5.6 to 9.2] | [8.8 to 12.9] | [12.3 to 19.1] | [20.4 to 28.1] | [20.7 to 42.0] | [7.4 to 9.7] |
| **Pain/Discomfort** | | | | | | | | |
| All respondents (%) | 6 | 9.2 | 13.4 | 18.6 | 20.4 | 27.1 | 30.3 | 14.3 |
| 95% CI | [4.4 to 8.1] | [7.7 to 10.9] | [11.5 to 15.5] | [16.6 to 20.7] | [17.9 to 23.1] | [23.3 to 31.2] | [24.1 to 37.2] | [13.0 to 15.6] |
| Male (%) | 3.6 | 5.5 | 7.6 | 10.6 | 14.1 | 16.5 | 28 | 8.9 |
| 95% CI | [2.3 to 5.7] | [4.0 to 7.3] | [6.0 to 9.6] | [9.1 to 12.4] | [11.4 to 17.4] | [13.1 to 20.7] | [20.7 to 36.7] | [8.0 to 9.9] |
| Female (%) | 8.5 | 11.9 | 18.2 | 25.9 | 27.2 | 41.2 | 35.1 | 19.1 |
| 95% CI | [5.7 to 12.3] | [9.8 to 14.5] | [15.5 to 21.3] | [23.0 to 29.1] | [23.5 to 31.1] | [35.6 to 47.0] | [25.0 to 46.7] | [17.1 to 21.2] |
| **Anxiety/Depression** | | | | | | | | |
| All respondents (%) | 4.9 | 5.8 | 7.6 | 9.9 | 10.9 | 13.7 | 18 | 8.1 |
| 95% CI | [3.7 to 6.6] | [4.8 to 7.0] | [6.5 to 8.9] | [8.7 to 11.4] | [9.4 to 12.7] | [11.2 to 16.7] | [13.3 to 23.7] | [7.4 to 8.9] |

Continued

**Table 1** Continued

| EQ5D dimensions | 20–24 years | 25–34 years | 35–44 years | 45–54 years | 55–64 years | 65–74 years | ≥75 years | Overall |
|---|---|---|---|---|---|---|---|---|
| Male (%) | 3.7 | 4.1 | 5.4 | 6.7 | 8.7 | 7.6 | 15.7 | 5.9 |
| 95% CI | [2.2 to 6.0] | [3.1 to 5.5] | [4.3 to 6.8] | [5.2 to 8.5] | [6.8 to 11.0] | [5.4 to 10.7] | [10.7 to 22.6] | [5.2 to 6.7] |
| Female (%) | 6.2 | 7.0 | 9.5 | 13 | 13.4 | 21.9 | 22.7 | 10.1 |
| 95% CI | [4.4 to 8.6] | [5.6 to 8.8] | [7.9 to 11.4] | [11.0 to 15.2] | [11.1 to 16.1] | [18.0 to 26.5] | [14.0 to 34.6] | [9.1 to 11.3] |

EQ5D, European Quality of Life Five Dimension.

### HRQOL and chronic conditions

Overall, individuals with chronic conditions reported lower health status than those without chronic conditions. About half of the respondents with self-reported diabetes, hypertension, stroke, heart disease or chronic kidney disease reported moderate or severe problems in all five domains (table 2).

Table 3 presents the adjusted prevalence ratio of moderate or severe problems among people with versus without chronic conditions, stratified by sex and cities. Individuals with chronic conditions reported two times greater problems in mobility, usual activities domains, pain/discomfort and anxiety/depression, than in individuals without chronic conditions.

Further, a small proportion of individuals with chronic conditions, mostly those with hypertension (10.5%) or diabetes (8.3%), reported having a perfect health state.

### Relationship between VAS and EQ5D measures across major subgroups

We expected that each EQ5D dimension would have an independent relationship with VAS since each of them represents a different aspect of HRQOL. Online supplementary appendix 2 provides the beta coefficients of the weighted regression models (ie, with the application of the population sampling weights). In the overall population, having any problems in mobility, self-care, pain/discomfort and anxiety/depression were associated with VAS scores that were 10–12 points lower. This inverse relationship of lower VAS with higher domain difficulties was larger in men, elderly (>60 years), low-income, less educated, divorced and high BMI individuals, compared with their respective counterparts. Tobacco users who reported difficulties in all domains of EQ5D had lower VAS scores (indicating lower quality of life). Kidney disease and stroke were the most disabling conditions on all measures.

### DISCUSSION

Comparative assessments of HRQOL variations by socio-demographic factors and chronic conditions aid in prioritising public health targets for intervention. Results from this study indicate that less than 10% of the respondents rated their health status as 100 (ie, best-imagined health state) on VAS. Mobility, pain/discomfort and anxiety/depression were the most commonly reported problems, with the extent of these problems differing across population subgroup. Elderly (>60 years) and women reported significantly greater problems in the mobility, pain/discomfort and anxiety/depression domains.

The mean VAS in our study was 74.5, which is lower than reported by most Western countries (82.5), but comparable to the results from other low-income and middle-income countries (LMIC) (71.1–77.8) (online supplementary appendix 3).[25–28] Lower health status reported by urban South Asians can be interpreted in a number of ways. The lower scores may be related

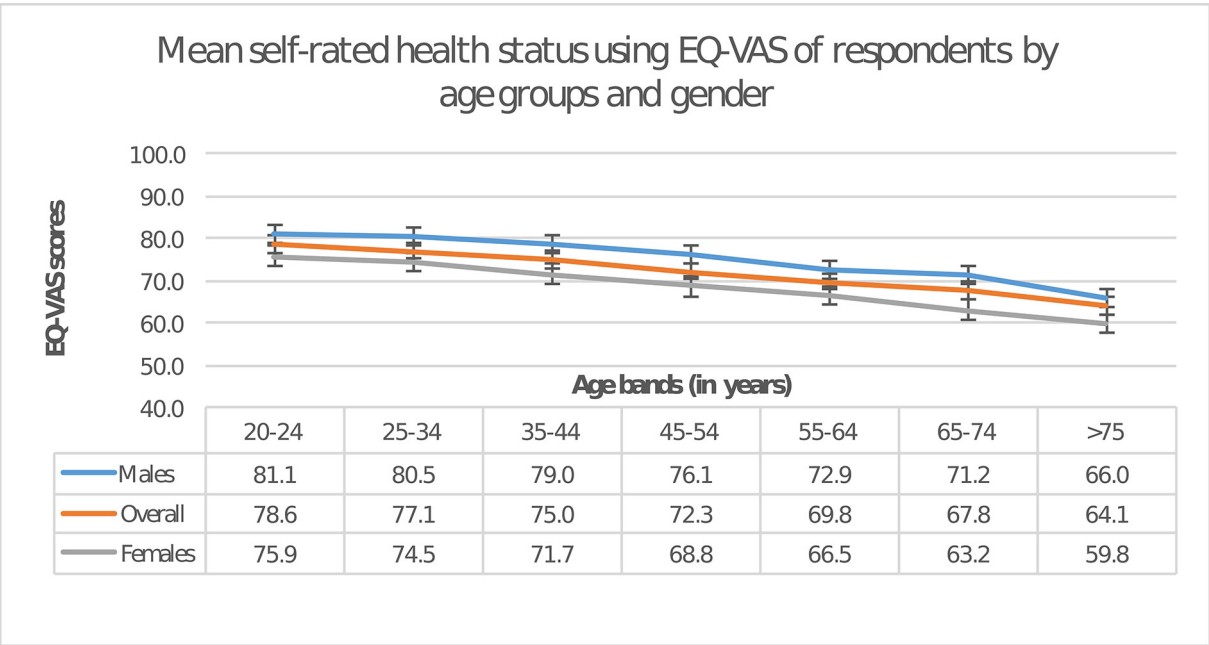

**Figure 1** Mean self-rated health status using the European Quality of Life Five Dimension—Visual Analogue Scale (EQ5D-VAS) of respondents by age groups and gender. This figure presents the mean self-rated health status for overall study population by age groups and gender. The EQ5D-VAS measures health status on a scale of 0 (worst health status) to 100 (best imaginable health status). *p Value for difference between mean EQ5D-VAS between men and women at each age group is statistically significant, p<0.01.

to generally lower reporting of HRQOL among this group. Alternatively, these scores may reflect morbidity and suboptimal access to healthcare facilities to address health concerns. South Asians experience chronic conditions at relatively younger ages than other race/ethnic groups, and the associated reductions in productivity and income levels may be manifested in these self-reported EQ5D-VAS scores.

A higher percentage of individuals reported problems in mobility dimension (14%–17%) in South Asia, which is comparable to results from other LMICs.[29] However, problems in anxiety/depression are pretty low in South Asia when compared with the rest of the world; this could be due to underdiagnoses of depression-related problems or stigma attached to these health conditions. These patterns could also reflect differences in how adults in different parts of the world self-rate their health.

Worse health status in retired or home makers, compared with employed persons, may be related to being homebound or reflect underlying illnesses that may be the factor driving these participants to be homebound and not employed.

In terms of modifiable risk factors, maintaining a healthy BMI cut-off (18–25 kg/m$^2$) is favoured because individuals with BMI <18 kg/m$^2$ and ≥25 kg/m$^2$ reported greater problems in all five domains. Although previous studies have shown that lower levels (intensity) of tobacco use are linked with higher HRQOL and regular tobacco users with worse health status,[30 31] in our study, former tobacco users reported lower HRQOL than current users. This finding may indicate reverse causality, that is, former

tobacco users after experiencing an illness would have quit smoking/tobacco. Further, supported by the fact that tobacco users with chronic conditions or greater difficulties in EQ5D domains had lower VAS scores is suggestive that morbidity and not the habit of tobacco use per se is more closely related to participants' perception of health. However, a causal link between tobacco use and HRQOL cannot be confirmed in this cross-sectional study. Longitudinal analyses of the independent associations between the smoking/tobacco with HRQOL may provide a better understanding of this relationship.

Notably, one in five individuals living with known hypertension or diabetes (average disease duration 4 years) still reported a perfect health state, indicating that these individuals may feel asymptomatic until they experience a clinical event. Also, very small proportions of patients with heart disease and stroke (with longer duration of illness; average 9 years) reported perfect health states, suggesting that these individuals may have adapted to their conditions over time and may be benefiting from treatment and self-care that improve their self-rated quality of life. However, we did not investigate whether these other factors like adherence influence quality of life in those living with chronic conditions.

Due to the differences in statistical analyses, HRQOL measures, sociodemographic characteristics of the sample and medical conditions selected, the results of this study may not be directly comparable to reports from other countries.[32] Nevertheless, a few differences and common findings are noteworthy. Individuals with stroke or chronic kidney disease rated the lowest health status,

**Table 2** Mean EQ-VAS and percentages of respondents reporting moderate or severe problems by various subgroups

| | Respondents (n) | EQ-VAS | | Mobility | | Self-care | | Usual activities | | Pain/Discomfort | | Anxiety/Depression | |
|---|---|---|---|---|---|---|---|---|---|---|---|---|---|
| | | Mean | 95% CI | % | 95% CI | % | 95% CI | % | 95% CI | % | 95% CI | % | 95% CI |
| **Cities** | | | | | | | | | | | | | |
| Chennai | 6906 | 70.7 | [70.1 to 71.4] | 17.3 | [15.4 to 19.4] | 7.8 | [6.5 to 9.2] | 7.7 | [6.6 to 9.0] | 10.3 | [9.1 to 11.7] | 11.4 | [9.8 to 13.1] |
| Delhi | 5364 | 78.8 | [77.8 to 79.8] | 14.1 | [11.9 to 16.7] | 1.6 | [1.2 to 2.1] | 4.4 | [3.4 to 5.7] | 19.4 | [16.9 to 22.2] | 5.2 | [4.0 to 6.6] |
| Karachi | 4017 | 73.2 | [72.6 to 73.8] | 10.4 | [9.4 to 11.5] | 1.9 | [1.5 to 2.4] | 5.6 | [4.9 to 6.4] | 12.9 | [11.8 to 14.2] | 7.2 | [6.3 to 8.1] |
| **Employment status** | | | | | | | | | | | | | |
| Employed | 7635 | 77.2 | [76.4 to 78.1] | 8.7 | [7.7 to 9.9] | 2.4 | [1.8 to 3.1] | 2.8 | [2.3 to 3.4] | 9.3 | [8.3 to 10.4] | 6.2 | [5.4 to 7.1] |
| Student | 361 | 77.5 | [75.2 to 79.7] | 6.5 | [4.1 to 10.0] | 1.5 | [0.7 to 3.4] | 2.6 | [1.3 to 4.9] | 8.3 | [5.4 to 12.7] | 7.0 | [4.4 to 10.8] |
| Home makers | 6781 | 71.6 | [70.7 to 72.4] | 20.9 | [18.9 to 22.9] | 5.7 | [4.8 to 6.8] | 9.0 | [7.8 to 10.4] | 19.7 | [17.6 to 22.0] | 9.9 | [8.9 to 11.0] |
| Retired | 765 | 71.6 | [69.9 to 73.3] | 21.0 | [17.7 to 24.6] | 7.7 | [5.7 to 10.3] | 11.0 | [8.5 to 14.0] | 18.1 | [15.0 to 21.8] | 9.0 | [7.0 to 11.5] |
| Unemployed | 743 | 68.3 | [66.8 to 69.9] | 17.1 | [13.6 to 21.2] | 7.7 | [5.7 to 10.5] | 9.5 | [7.3 to 12.4] | 16.1 | [13.2 to 19.6] | 12.3 | [9.8 to15.2] |
| **Income class** | | | | | | | | | | | | | |
| Low-income group (<10 000 Indian rupees or US$ 155) | 11 537 | 73.4 | [72.7 to 74.1] | 15.2 | [13.9 to 16.7] | 4.8 | [4.1 to 5.7] | 6.8 | [6.0 to 7.6] | 14 | [12.7 to 15.4] | 9.2 | [8.3 to 10.1] |
| Middle-income group (10000–20 000 Indian rupees or US$155–310) | 2667 | 75.8 | [74.8 to 76.9] | 14.6 | [12.6 to 16.8] | 3.5 | [2.6 to 4.7] | 4.9 | [3.9 to 6.1] | 14.5 | [12.4 to 17.0] | 6.4 | [5.3 to 7.6] |
| High-income group (>20 000 Indian rupeesor US$>310) | 1975 | 77.1 | [75.9 to 78.3] | 11.0 | [8.8 to 13.8] | 1.5 | [1.0 to 2.2] | 3.4 | [2.3 to 4.9] | 15.5 | [12.8 to 18.6] | 5.0 | [3.9 to 6.4] |
| **Education status** | | | | | | | | | | | | | |
| Up to primary school | 3604 | 71.1 | [70.1 to 72.2] | 21.6 | [19.5 to 24.0] | 5.6 | [4.7 to 6.7] | 9.8 | [8.5 to 11.2] | 20.8 | [18.5, 23.3] | 10.9 | [9.6 to 12.4] |
| Secondary school | 9924 | 74.3 | [73.5 to 75.1] | 14.0 | [12.6 to 15.5] | 4.5 | [3.7 to 5.3] | 5.8 | [5.1 to 6.7] | 13.0 | [11.8 to 14.4] | 8.1 | [7.3 to 9.0] |
| Graduation and above | 2759 | 77.9 | [76.9 to 78.8] | 8.3 | [6.7 to 10.2] | 1.6 | [1.1 to 2.2] | 2.3 | [1.7 to 3.1] | 10.8 | [9.0 to 12.8] | 5.0 | [4.1 to 6.0] |
| **Marital status** | | | | | | | | | | | | | |
| Single | 1177 | 78.2 | [76.9 to 79.6] | 6.7 | [5.2 to 8.7] | 1.6 | [0.9 to 2.9] | 2.0 | [1.2 to 3.2] | 7.1 | [5.5 to 9.2] | 5.4 | [3.9 to 7.5] |
| Married | 14217 | 74.3 | [73.6 to 75.0] | 14.2 | [13.0 to 15.6] | 4.1 | [3.5 to 4.8] | 5.7 | [5.0 to 6.5] | 14 | [12.6 to 15.4] | 7.8 | [7.1 to 8.6] |
| Widowed | 838 | 67.4 | [66.1 to 68.7] | 34.5 | [30.5 to 38.8] | 10.2 | [8.0 to 12.9] | 18.7 | [15.8 to 22.1] | 32.7 | [29.3 to 36.2] | 18.2 | [15.4 to 21.3] |
| Separated/Divorced | 55 | 65.3 | [57.9 to 72.7] | 24.6 | [11.8 to 44.3] | 10.2 | [4.6 to 21.3] | 20.6 | [8.8 to 41.0] | 22.3 | [12.2 to 37.2] | 25.1 | [14.9 to 39.2] |
| **BMI (kg/m²)** | | | | | | | | | | | | | |
| Underweight (<18) | 756 | 74.0 | [72.3 to 75.7] | 14.9 | [11.7 to 18.8] | 3.0 | [1.7 to 5.3] | 5.4 | [3.7 to 7.9] | 12.9 | [10.1 to 16.5] | 9.0 | [6.3 to 12.7] |
| Normal (18–24.9) | 5278 | 75.1 | [74.3 to 75.9] | 11.8 | [10.4 to 13.3] | 3.5 | [2.8 to 4.2] | 4.8 | [4.0 to 5.7] | 11.5 | [10.2 to 13.0] | 7.5 | [6.6 to 8.7] |
| Overweight (25.0–29.9) | 4190 | 73.6 | [72.7 to 74.5] | 15.2 | [13.4 to 17.2] | 4.8 | [3.8 to 6.0] | 6.2 | [5.2 to 7.3] | 14.4 | [12.6 to 16.4] | 8.2 | [7.2 to 9.4] |
| Obesity (≥30) | 2249 | 70.4 | [69.5 to 71.2] | 22.3 | [19.9 to 24.9] | 6.4 | [5.1 to 8.0] | 9.1 | [7.6 to 10.9] | 20.7 | [18.2 to 23.6] | 10 | [8.5 to 11.8] |
| **Tobacco use (smoke/chew/other forms)** | | | | | | | | | | | | | |
| Never user | 12215 | 74.1 | [73.4 to 74.8] | 15.4 | [14.0 to 16.8] | 4.5 | [3.8 to 5.3] | 6.4 | [5.6 to 7.2] | 14.6 | [13.1 to 16.2] | 8.1 | [7.3 to 9.0] |
| Current user | 3758 | 75.3 | [74.3 to 76.3] | 11.8 | [10.4 to 13.4] | 3.1 | [2.4 to4.1] | 4.6 | [3.8 to 5.6] | 12.9 | [11.6 to 14.5] | 7.9 | [6.7 to 9.2] |
| Former user | 314 | 70.1 | [67.0 to 73.2] | 17.6 | [13.4 to 22.8] | 4.7 | [2.7 to 8.1] | 8.3 | [5.6 to 12.3] | 18.1 | [13.6 to 23.7] | 12.4 | [8.8 to 17.2] |
| **Chronic conditions (self-reported)** | | | | | | | | | | | | | |

Continued

**Table 2** Continued

| | Respondents (n) | EQ-VAS | | Mobility | | Self-care | | Usual activities | | Pain/Discomfort | | Anxiety/Depression | |
|---|---|---|---|---|---|---|---|---|---|---|---|---|---|
| | | Mean | 95% CI | % | 95% CI | % | 95% CI | % | 95% CI | % | 95% CI | % | 95% CI |
| No | 12 498 | 76.2 | [75.4 to 76.9] | 11.9 | [10.7 to 13.1] | 3.5 | [2.8 to 4.2] | 4.5 | [3.9 to 5.2] | 11.2 | [10.1 to 12.5] | 6.8 | [6.0 to 7.6] |
| Yes | 4699 | 67.3 | [66.6 to 68.1] | 24.6 | [22.4 to 27.0] | 6.9 | [5.9 to 7.9] | 11.8 | [10.5 to 13.2] | 25.5 | [23.3 to 27.9] | 13.2 | [12.0 to 14.5] |
| Diabetes | | | | | | | | | | | | | |
| No diabetes | 4610 | 75.1 | [74.4 to 75.9] | 10.1 | [8.9 to 11.4] | 3.3 | [2.6 to 4.1] | 4.2 | [3.5 to 5.0] | 10.3 | [8.8 to 12.0] | 7.5 | [6.5 to 8.6] |
| Pre-diabetes | 5449 | 74.4 | [73.5 to 75.3] | 15.5 | [13.6 to 17.5] | 4.2 | [3.3 to 5.2] | 6.0 | [5.0 to 7.2] | 14.6 | [12.9 to 16.4] | 8.0 | [7.0 to 9.1] |
| Newly diagnosed | 2015 | 74.3 | [73.3 to 75.3] | 17.2 | [14.7 to 20.1] | 5.0 | [3.8 to 6.5] | 8.1 | [6.5 to 10.0] | 17.2 | [15.0 to 19.6] | 8.5 | [7.1 to 10.1] |
| Self-reported diabetes | 1661 | 65.9 | [64.8 to 67.1] | 20.9 | [18.9 to 23.1] | 6.3 | [5.2 to 7.5] | 9.5 | [8.3 to 11.0] | 19.8 | [18.1 to 21.7] | 10.5 | [9.3 to 11.8] |
| Hypertension | | | | | | | | | | | | | |
| Normotension | 5695 | 74.8 | [74.0 to 75.5] | 12.8 | [11.4 to 14.3] | 3.8 | [3.0 to 4.9] | 4.7 | [3.9 to 5.6] | 11.6 | [9.9 to 13.6] | 6.9 | [6.0 to 8.0] |
| Prehypertension | 4717 | 76.0 | [75.0 to 76.9] | 12.7 | [11.2 to 14.4] | 3.7 | [2.9 to 4.6] | 5.1 | [4.3 to 6.0] | 12.3 | [11.0 to 13.8] | 6.5 | [5.6 to 7.5] |
| Newly diagnosed | 2780 | 75.8 | [74.8 to 76.8] | 12.9 | [11.0 to 15.1] | 3.0 | [2.4 to 3.8] | 5.1 | [4.2 to 6.3] | 12.6 | [11.0 to 14.3] | 8.4 | [7.2 to 9.9] |
| Self-reported | 2397 | 66.7 | [65.8 to 67.7] | 18.8 | [16.8 to 21.0] | 4.9 | [4.2 to 5.7] | 8.7 | [7.6 to 9.9] | 19.6 | [17.8 to 21.4] | 10.8 | [9.7 to 12.0] |
| Heart disease | | | | | | | | | | | | | |
| No | 15 842 | 74.6 | [73.9 to 75.2] | 14.2 | [12.9 to 15.5] | 4.0 | [3.4 to 4.7] | 5.6 | [5.0 to 6.4] | 13.8 | [12.6 to 15.2] | 7.9 | [7.2 to 8.6] |
| Yes | 445 | 63.3 | [61.3 to 65.2] | 31.2 | [25.7 to 37.3] | 11.7 | [8.3 to 16.2] | 20.8 | [16.7 to 25.7] | 31.4 | [26.5 to 36.7] | 19.0 | [15.0 to 23.8] |
| Stroke | | | | | | | | | | | | | |
| No | 16 203 | 74.3 | [73.6 to 75.1] | 14.5 | [13.3 to 15.8] | 4.1 | [3.5 to 4.8] | 6.0 | [5.3 to 6.7] | 14.1 | [12.9 to 15.5] | 8.1 | [7.4 to 8.8] |
| Yes | 84 | 62.4 | [58.5 to 66.2] | 31.9 | [22.0 to 43.7] | 16.8 | [10.0 to 26.7] | 18.1 | [11.1 to 28.1] | 43.3 | [32.3 to 55.1] | 21.2 | [13.2 to 32.2] |
| Kidney disease | | | | | | | | | | | | | |
| No | 16 175 | 74.4 | [73.7 to 75.1] | 14.6 | [13.3 to 15.9] | 4.1 | [3.6 to 4.8] | 6.0 | [5.3 to 6.7] | 14.1 | [12.9 to 15.5] | 8.1 | [7.3 to 8.8] |
| Yes | 112 | 62.5 | [59.4 to 65.7] | 19.2 | [13.1 to 27.2] | 10 | [5.7 to 17.0] | 15.1 | [9.9 to 22.3] | 31.4 | [23.5 to 40.6] | 20.7 | [14.1 to 29.3] |

Tobacco use, heart disease and kidney disease were based on self-reports; newly diagnosed diabetes, defined as no self-reported diabetes and fasting blood glucose (FBG) of ≥126 mg/dL or HbA1c ≥6.5%; pre-diabetes, no self-reported diabetes and FBG ≥100–125 mg/dL or HbA1c ≥5.7%–6.4%; normoglycaemia, no self-reported diabetes and FBG <100 mg/dL and HbA1c <5.7%; newly diagnosed hypertension, defined as no self-reported hypertension and blood pressure ≥140/90 mm Hg; prehypertension, no self-reported hypertension and blood pressure 120–139/80–89 mm Hg; and normotensive, no history of hypertension and blood pressure <120/80 mm Hg.
BMI, body mass index; EQ-VAS, European Quality of Life—Visual Analogue Scale; HbA1c, glycated haemoglobin.

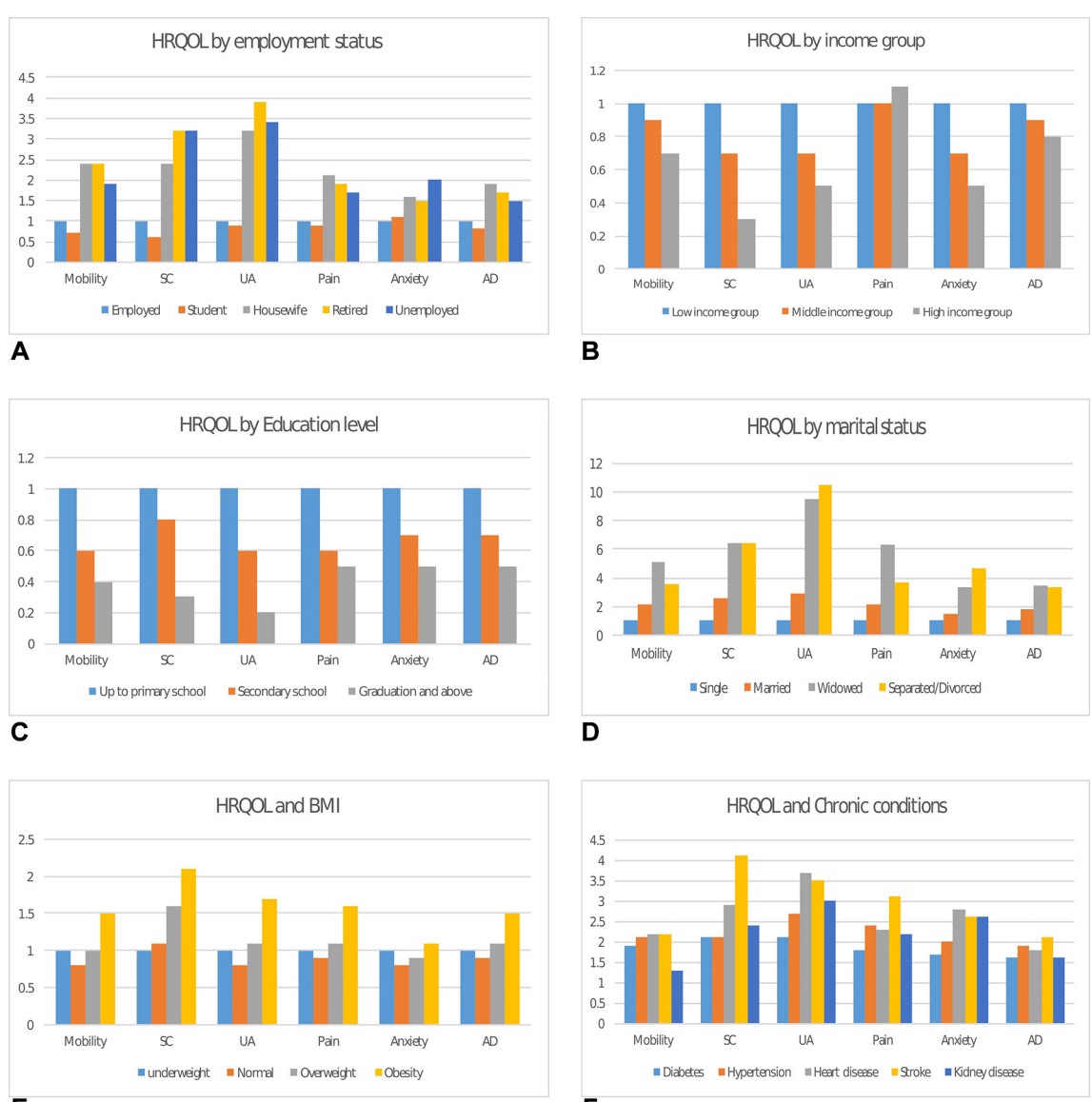

**Figure 2** PR of moderate or severe health problems by sociodemographic factors and chronic conditions. (A) Shows the PR of moderate or severe difficulties in EQ5D domains (mobility, self-care, usual activities, pain/discomfort, anxiety/depression and any of the five dimensions) by employment status. With reference to those who were employed (PR=1), housewife, retired and the unemployed reported greater problems in all five domains, whereas students only reported higher anxiety problems compared with employed. (B) Shows the PR of moderate or severe difficulties in EQ5D domains by income group. With reference to low-income group (PR=1), those in middle-income or high-income groups had less problems in all five domains. (C) Shows the PR of moderate or severe difficulties in EQ5D domains (mobility, self-care, usual activities, pain/discomfort, anxiety/depression and any of the five dimensions) by education level. With reference to those primary school education (PR=1), individuals with secondary school or graduates reported significantly lower problems in all five domains. (D) Shows the PR of moderate or severe difficulties in EQ5D domains by marital status. With reference to single (PR=1), those who were married, widower or divorcee had greater problems in all five domains. (E) Shows the PR of moderate or severe difficulties in EQ5D domains (mobility, self-care, usual activities, pain/discomfort, anxiety/depression and any of the five dimensions) by BMI. With reference to underweight, that is, BMI <18 kg/m$^2$ (PR=1), individuals who are overweight (BMI 25–29.9 kg/m$^2$) or obese (BMI ≥30 kg/m$^2$) reported significantly greater problems in all five domains. (F) Shows the PR of moderate or severe difficulties in EQ5D domains by chronic conditions. Compared with those without chronic conditions, individuals with self-reported diabetes, hypertension, heart disease, stroke and kidney disease had twice greater problems in all five domains. AD, any dimension; BMI, body mass index; EQ5D, European Quality of Life Five Dimension; HRQOL, health-related quality of life; PR, prevalence ratio; SC, self-care; UA, usual activities.

which is consistent with results reported from other studies done in China, Thailand and Western populations.[28][33–36] Since the respondent's health status could be affected by how well the condition was managed, caution is needed in interpreting study results regarding the relative effect of chronic conditions on HRQOL.[37–41] A more

Table 3  Adjusted PR of reporting any problems in individuals with chronic conditions versus those without chronic conditions, by cities and gender

| | Mobility | | Self-care | | Usual activities | | Pain/Discomfort | | Anxiety/Depression | | Any dimension | |
|---|---|---|---|---|---|---|---|---|---|---|---|---|
| | PR | 95% CI | PR | 95% CI | PR | 95% CI | PR | 95% CI | PR | 95% CI | PR | 95% CI |
| Overall* | 1.41 | [1.27 to 1.57] | 1.35 | [1.09 to 1.67] | 1.70 | [1.45 to 2.01] | 1.74 | [1.58 to 1.91] | 1.60 | [1.40 to 1.84] | 1.45 | [1.35 to 1.56] |
| Male† | 1.78 | [1.46 to 2.18] | 1.60 | [1.10 to 2.34] | 2.01 | [1.41 to 2.87] | 1.89 | [1.62 to 2.23] | 2.01 | [1.55 to 2.62] | 1.63 | [1.41 to 1.88] |
| Female† | 1.32 | [1.16 to 1.50] | 1.28 | [0.99 to 1.66] | 1.67 | [1.40 to 1.99] | 1.72 | [1.53 to 1.93] | 1.44 | [1.23 to 1.69] | 1.39 | [1.29 to 1.52] |
| Chennai‡ | 1.24 | [1.05 to 1.46] | 1.08 | [0.84 to 1.40] | 1.31 | [1.07 to 1.60] | 1.71 | [1.45 to 2.04] | 1.56 | [1.31 to 1.86] | 1.30 | [1.17 to 1.44] |
| Male† | 1.46 | [1.07 to 1.99] | 1.12 | [0.71 to 1.76] | 1.24 | [0.80 to 1.90] | 1.73 | [1.34 to 2.24] | 1.75 | [1.34 to 2.30] | 1.38 | [1.16 to 1.66] |
| Female† | 1.18 | [0.97 to 1.44] | 1.08 | [0.79 to 1.47] | 1.38 | [1.11 to 1.73] | 1.73 | [1.38 to 2.17] | 1.46 | [1.16 to 1.83] | 1.27 | [1.12 to 1.45] |
| Delhi‡ | 1.66 | [1.41 to 1.95] | 3.05 | [1.78 to 5.25] | 2.40 | [1.66 to 3.47] | 1.79 | [1.54 to 2.08] | 1.84 | [1.29 to 2.63] | 1.65 | [1.47 to 1.86] |
| Male† | 2.13 | [1.52 to 2.97] | 5.14 | [1.92 to 13.78] | 3.63 | [1.61 to 8.17] | 1.93 | [1.50 to 2.49] | 3.45 | [1.36 to 8.80] | 1.89 | [1.45 to 2.48] |
| Female† | 1.58 | [1.31 to 1.90] | 2.23 | [1.27 to 3.94] | 2.15 | [2.15 to 2.15] | 1.82 | [1.51 to 2.18] | 1.56 | [1.12 to 2.18] | 1.62 | [1.43 to 1.83] |
| Karachi‡ | 1.41 | [1.15 to 1.72] | 1.77 | [1.05 to 3.00] | 2.09 | [1.58 to 2.77] | 1.64 | [1.37 to 1.97] | 1.60 | [1.21 to 2.11] | 1.43 | [1.25 to 1.65] |
| Male† | 2.51 | [1.52 to 4.13] | 1.76 | [0.77 to 4.02] | 2.91 | [1.62 to 5.22] | 2.28 | [1.52 to 3.42] | 2.04 | [1.27 to 3.29] | 2.01 | [1.44 to 2.80] |
| Female† | 1.24 | [1.01 to 1.53] | 2.11 | [1.10 to 4.04] | 1.95 | [1.44 to 2.65] | 1.50 | [1.24 to 1.81] | 1.32 | [0.94 to 1.87] | 1.27 | [1.10 to 1.47] |

Prevalence ratios (PRs) were estimated using Poisson regression model as the log binomial regression model did not reach convergence.

*Adjusted for age, sex, income, education, marital status and city.
†Adjusted for age, income, education, marital status and city.
‡Adjusted for age, sex, income, education and marital status.

recent Canadian study conducted by Mo *et al*[42] indicated a strong relationship between low Health Utility Index scores and certain chronic conditions. The authors found that arthritis/rheumatism, heart disease, hypertension, cataracts and diabetes had a negative impact on HRQOL. In the USA, Medical Expenditure Panel Survey data-based study reported that, after adjusting for sociodemographic variables, all of the selected chronic conditions were associated with lower EQ5D scores, with effects greatest for emphysema, followed by heart disease, stroke, high BP, diabetes and asthma.[43 44]

## Strengths and limitations of this study

To our knowledge, this is the first population-level HRQOL data from South Asia using EQ5D-VAS including three large metropolitan cities in India and Pakistan with a large sample size that has used multistage cluster random sampling strategy and standardised protocols and measurement tools across sites. Our data provide the first baseline values to be used for monitoring population health status and analysed the relationships between selected chronic conditions and HRQOL. This information could be used to complement national targets by providing a measure of chronic disease burden based on perceived health status rather than solely on mortality and disease prevalence. In our secondary data analysis, EQ5D and VAS measures correlated well, which confirms the convergent and discriminate validity of the EQ5D instrument.

There are several limitations to this study. First, due to the cross-sectional nature of the data, the causal relationship between socioeconomic parameters/chronic conditions and HRQOL cannot be determined and is not implied. Second, many chronic conditions (respiratory, locomotor, cancer and others) were not included in the survey. Therefore, the ranking of most severe health conditions and associated HRQOL is not complete. Third, the selected chronic conditions were self-reported, and the study investigators did not examine the accuracy of information. However, this poses less of a threat to validity because self-reporting of heart diseases, stroke and kidney diseases is pretty accurate in community surveys.[45–48] Further, hypertension and diabetes were measured in this study using standardised methods. Lastly, EQ5D data were self-reported and the variation in how individuals perceive disability varies widely. However, this should be less of a problem given the large sample size in this study. Fourth, the findings of this study may not be replicable if researchers use a different HRQOL instrument,[49–54] which can be tested in a future study.

## Public health relevance and policy implications

HRQOL data from this study provide baseline values for monitoring variations in health for specific population groups on the basis of gender, education, employment, income, presence of chronic conditions and place of residence. These data are also relevant to assess the overall burden of physical and mental health problems that are not disease-specific. In aggregate form, such information could be used to complement national health targets by providing a measure based on health status (quality of life) rather than mortality or disease prevalence alone. Therefore, the policy makers can use the HRQOL measures and the resulting data from this study to minimise health disparities and allocate resources among competing health programme based on burden of physical or mental health problems in a specific group.[55]

The lower health status reported by female, less educated, unemployed and low-income groups may indicate higher levels of stress in these groups.[17] Other potential contributing factors that are known to influence health status are living conditions, gross domestic product per capita, inequities in income distributions and access to healthcare.[56–61] Therefore, public health initiatives should focus on intersectoral approaches to address issues of education, generating more avenues for employment and improving the quality and access of primary healthcare.

Lastly, the issue of 'clinical' or 'policy' relevance of the difference in EQ5D measures needs much discourse. For example, if the VAS in two groups of the population is 5 or 10 points different from each other, we cannot make a clinical judgement on how much these two groups would differ in their actual health status. These issues relate to determining a minimally significant difference/change in HRQOL and needs investigation in future studies. However, because of HRQOL sensitivity to time trends as shown in previous studies,[62–64] these measures are also likely to be useful in determining the effect of major population-based policies or interventions.

## CONCLUSION

HRQOL appears to be lower with higher age and among women in South Asia. Our data demonstrate significantly lower HRQOL in key demographic groups and those with chronic conditions, which are consistent with previous studies. These data provide insights on inequalities in population health status, and potentially reveal unmet needs in the community to guide health policies.

**Author affiliations**
[1]Centre for Control of Chronic Conditions (4C), New Delhi, India
[2]Public Health Foundation of India, New Delhi, India
[3]Centre for Chronic Disease Control, New Delhi, India
[4]Department of Endocrinology and Metabolism, All India Institute of Medical Sciences, New Delhi, Delhi, India
[5]Hubert Department of Global Health, Emory University, Atlanta, Georgia, USA
[6]Department of Epidemiology, Rollins School of Public Health, Emory University, Atlanta, Georgia, USA
[7]Madras Diabetes Research Foundation and, Dr. Mohan's Diabetes Specialties Centre, Chennai, Tamil Nadu, India
[8]Department of Community Health Sciences, Aga Khan University, Karachi, Pakistan
[9]School of Medicine, University of Washington, Seattle, Washington, USA
[10]London School of Hygiene and Tropical Medicine, London, United Kingdom

**Contributors** KS, DP, MKA, RS, NT and KMVN conceptualised and designed the study. KS wrote the first draft of the manuscript. KS and DK performed statistical analysis. RS, VSA, MKA, RP, VM, MMK, MDS, NT, KMVN and DP contributed

significantly in the revision of the manuscript. All authors have approved the submission of this version of the manuscript.

**Funding** The CARRS Study was funded in whole or in part by the National Heart, Lung, and Blood Institute of the National Institutes of Health, Department of Health and Human Services (contract no HHSN268200900026C) and the United Health Group (Minneapolis, Minnesota, USA). RS is supported by a Wellcome Trust Capacity Strengthening Strategic Award Extension phase to the Public Health Foundation of India and a consortium of UK universities (WT084754/Z/08/A).

**Competing interests** None declared.

**Ethics approval** The CARRS study has obtained institutional ethics approval from each of the participating institutions: Public Health Foundation of India, All India Institute of Medical Sciences, New Delhi, India; Madras Diabetes Research Foundation, Chennai, India; Aga Khan University, Karachi; Rollins School of Public Health, Emory University, USA.

**Provenance and peer review** Not commissioned; externally peer reviewed.

**Data sharing statement** KS, DK and DP have access to study data set and statistical code. Any request for data sharing should be addressed to the corresponding author (DP).

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
