## [Reviewer comments · BMJ Open]

ARTICLE DETAILS

TITLE (PROVISIONAL)	Health-related Quality of Life Variations by Socio-demographic Factors and Chronic Conditions in three metropolitan cities of South Asia: The CARRS Study
AUTHORS	Singh, Kavita; Kondal, Dimple; Shivashankar, Roopa; Ali, Mohammed; Pradeepa, Rajendra; Vamadevan, Ajay; Mohan, V; Kadir, Muhammad; Sullivan, Mark; Tandon, Nikhil; Narayan, K; Prabhakaran, Dorairaj

VERSION 1 – REVIEW

REVIEWER	Bishwajit Ghose Tongji Medical College, China
REVIEW RETURNED	29-Jul-2017

GENERAL COMMENTS	The study addresses an important research gap in this area in the context of three major cities in India and Pakistan combined. Authors might want to rewrite the title since the sample population does not represent South Asia. page 6 line 44: did the authors measure VAS for developed countries themselves, if not than it should be removed and discussed later. page 8 line 26: is the term 'fear' necessary here? Authors should justify the use of international cut-off for BMI instead of the Asian one. On what basis income was categorised into those three standards. Are the standards same for both of the countries. On a number of occasions readers encounter substandard sentencing and require revision, e.g. Mobility is the dimension that has the highest rate of having problems... Please consider providing some policy implications of the present study.
---

REVIEWER	Dr. Ravi Samuel The Psychotherapy Clinic, India
REVIEW RETURNED	08-Aug-2017

GENERAL COMMENTS	I wish to congratulate the authors for an outstanding work on Quality of Life of people with different ailments. The presentation of the
--

	article reflects the research thoroughness in which the work has been carried out. Explaining even a minor detail; why anxiety and depression is felt less in India and Pakistan is very impressive It was a delight to read the article.
--	---

REVIEWER	Antonio Bernabe-Ortiz Universidad Peruana Cayetano Heredia, Lima, Peru
REVIEW RETURNED	08-Aug-2017

GENERAL COMMENTS	Although this is a good piece of work, my main concern is regarding the use of logistic regression models to report OR instead of reporting prevalence ratios as in cross-sectional studies. Given that the prevalence of different domains of the EQ5D is over 10%, I suggest including PR in the results section. Methods Page 11: Good to add an equivalence of currency in USD for international readers (instead of adding it in the Results) Page 11: Please, define appropriately BMI cutoffs (BMI ≥ 25 is overweight and not normal as pointed out in the manuscript). Unless, the authors use another categorization (in that case please add the reference). Page 12: Being this a cross sectional study, it would be better to report prevalence ratios instead of OR as suggested by Barros and Hirakata. Please, make this change through the document. Results Page 13: "A third (66%) of the study population...". A third of the population is 33% and not 66%. Please correct. Page 13: "Women reported lower health status than men...": Good if a p-value can be added in the comparison. Discussion It would be good if a paragraph regarding the public health relevance of the findings is added to the manuscript. Much of the discussion is centered on comparison with other studies but not the utility of the results.
--

VERSION 1 – AUTHOR RESPONSE

Reviewer: 1

Reviewer Name: Bishwajit Ghose

Institution and Country: Tongji Medical College, China Please state any competing interests or state

'None declared': None

Please leave your comments for the authors below

1. The study addresses an important research gap in this area in the context of three major cities in India and Pakistan combined. Authors might want to rewrite the title since the sample population does not represent South Asia.

Response: As suggested, we have modified the title as "Health-related Quality of Life Variations by Socio-demographic Factors and Chronic Conditions in three metropolitan cities of South Asia: The CARRS Study" in the revised manuscript. (Refer page no. 1; line: 3).

2. page 6 line 44: did the authors measure VAS for developed countries themselves, if not than it should be removed and discussed later.

Response: No. We did not measure VAS in developed countries. We have deleted this line from the conclusion in the revised abstract to avoid confusion. (Refer page no. 6; lines: 19-20.)

3. page 8 line 26: is the term 'fear' necessary here?

Response: Thank you for this query. The fear of hypoglycaemia is a real one. Many patients with diabetes taking insulin or other oral hypoglycaemic agents, do experience anxiety/depression and have real fear of hypoglycaemia. So, we believe, it is important to emphasize on this fact along with the episodes of hypoglycaemia and other recurrent events and complications.

4. Authors should justify the use of international cut-off for BMI instead of the Asian one.

Response: Thank you for this query. We had analysed our data using both international cut-off and Asian criteria for BMI categories and results were comparable. We preferred reporting international cut-off as this would make comparison with other studies easier. We hope this fine.

5. On what basis income was categorised into those three standards. Are the standards same for both of the countries?

Response: Thank you for this query. To clarify, monthly income categories: (less than Indian rupees (INR)10,000 (equivalent to US\$200), INR10,000– 20,000 (US\$200–400) and greater than INR 20,000 (US\$400)) has been reported in previously published papers from the CARRS-Surveillance Study. To be consistent with previous reports from the CARRS-Study we have used the same income categories, again this would make comparison with other studies easier. Yes, same classification was applied to both the countries data.

6. On a number of occasions readers encounter substandard sentencing and require revision, e.g. Mobility is the dimension that has the highest rate of having problems...

Response: Thank you for this comment. We have reviewed the whole manuscript and improved the sentences for clarity.

7. Please consider providing some policy implications of the present study.

Response: As suggested by the reviewer, we have included a brief section on policy implications of the present study. (Refer page number 20; lines 2-25; and page no. 21; lines: 1-2).

Reviewer: 2

Reviewer Name: Dr. Ravi Samuel

Institution and Country: The Psychotherapy Clinic, India Please state any competing interests or state 'None declared': None Declared

Please leave your comments for the authors below

Comment; I wish to congratulate the authors for an outstanding work on Quality of Life of people with different ailments. The presentation of the article reflects the research thoroughness in which the work

has been carried out. Explaining even a minor detail; why anxiety and depression is felt less in India and Pakistan is very impressive

It was a delight to read the article.

Response: Thank you very much. We appreciate the reviewer's enthusiasm for our paper.

Reviewer: 3

Reviewer Name: Antonio Bernabe-Ortiz

Institution and Country: Universidad Peruana Cayetano Heredia, Lima, Peru Please state any competing interests or state 'None declared': None declared

Please leave your comments for the authors below

Although this is a good piece of work, my main concern is regarding the use of logistic regression models to report OR instead of reporting prevalence ratios as in cross-sectional studies. Given that the prevalence of different domains of the EQ5D is over 10%, I suggest including PR in the results section.

Response: Thank you for this insightful suggestion. We have now reported prevalence ratios in the results section. Refer Table 3 and Figure 2.a – 2.f.

VERSION 2 – REVIEW

REVIEWER	Antonio Bernabe-Ortiz Universidad Peruana Cayetano Heredia, Lima, Peru.
REVIEW RETURNED	11-Sep-2017
GENERAL COMMENTS	Page 21, Line 10: Please, correct "...resulting data from...".. instead "...resulting data form"